# AI Enabled Accident Detection and Alert System Using IoT and Deep Learning for Smart Cities

Nikhlesh Pathik [1], Rajeev Kumar Gupta [2,*], Yatendra Sahu [3], Ashutosh Sharma [4,5,*], Mehedi Masud [6] and Mohammed Baz [7]

1   Computer Science and Engineering, Sagar Institute of Science & Technology, Bhopal 462030, India; nikhleshpathik@sistec.ac.in
2   Computer Science and Engineering, Pandit Deendayal Petroleum University, Gandhinagar 382007, India
3   Computer Science and Engineering Indian Institute of Information Technology, Bhopal 462003, India; cse0016@iiitbhopal.ac.in
4   Institute of Computer Technology and Information Security, Southern Federal University, 344006 Rostov Oblast, Russia
5   School of Computer Science, University of Petroleum and Energy Studies, Dehradun 248171, India
6   Department of Computer Science, College of Computers and Information Technology, Taif University, P.O. Box 11099, Taif 21944, Saudi Arabia; mmasud@tu.edu.sa
7   Department of Computer Engineering, College of Computers and Information Technology, Taif University, P.O. Box 11099, Taif 21944, Saudi Arabia; mo.baz@tu.edu.sa
*   Correspondence: rajeev.gupta@sot.pdpu.ac.in (R.K.G.); sharmaashutosh1326@gmail.com (A.S.)

**Abstract:** As the number of vehicles increases, road accidents are on the rise every day. According to the World Health Organization (WHO) survey, 1.4 million people have died, and 50 million people have been injured worldwide every year. The key cause of death is the unavailability of medical care at the accident site or the high response time in the rescue operation. A cognitive agent-based collision detection smart accident alert and rescue system will help us to minimize delays in a rescue operation that could save many lives. With the growing popularity of smart cities, intelligent transportation systems (ITS) are drawing major interest in academia and business, and are considered as a means to improve road safety in smart cities. This article proposed an intelligent accident detection and rescue system which mimics the cognitive functions of the human mind using the Internet of Things (IoTs) and the Artificial Intelligence system (AI). An IoT kit is developed that detects the accident and collects all accident-related information, such as position, pressure, gravitational force, speed, etc., and sends it to the cloud. In the cloud, once the accident is detected, a deep learning (DL) model is used to validate the output of the IoT module and activate the rescue module. Once the accident is detected by the DL module, all the closest emergency services such as the hospital, police station, mechanics, etc., are notified. Ensemble transfer learning with dynamic weights is used to minimize the false detection rate. Due to the dataset's unavailability, a personalized dataset is generated from the various videos available on the Internet. The proposed method is validated by a comparative analysis of ResNet and InceptionResnetV2. The experiment results show that InceptionResnetV2 provides a better performance compared to ResNet with training, validation, and a test accuracy of 98%, respectively. To measure the performance of the proposed approach in the real world, it is validated on the toy car.

**Keywords:** AI; intelligent transportation systems (ITS); cognitive science; deep learning; IoT; ResNet; InceptionResnetV2; accident detection; sensors

## 1. Introduction

The demands for vehicles are increasing exponentially as the population increases. The percentage of road accidents has grown tremendously in the last few years, which is an alarming situation for everyone. According to a detailed analysis undertaken by the WHO,

road accidents claim the lives of millions of people each year and are the world's eighth largest cause of death. Road accidents are expected to become the fifth greatest cause of mortality in the future, according to the Association for Safe International Road Travel (ASIRT) [1]. The ASIRT reports that each country spends between one and two percent of its annual budget on road accidents [2]. The Transport Research Wing (TRW) of India surveyed road accidents in India and claimed that nearly 1230 accidents and 414 deaths were reported every day in 2019. According to this survey, 51 accidents and 17 deaths were reported every hour in 2019 [2]. Nowadays, people's lives are compromised on the roads due to their careless conduct and flouting of traffic laws on their whims and fancies. There are multiple causes of an accident such as high speed, overtaking, using a mobile phone, weather conditions, etc.

According to a recent study [2,3], over speeding is the main cause of accidents. In the rescue operation, the location of the accident spot is important [4,5]. In the case of heavy traffic or a city location, emergency assistance will be available shortly, but in low-traffic areas or highways, it is difficult to provide emergency aid on time. It is observed that significant injuries are converted into death because of delays in medical care. The victim's survival rate depends very much on how long an ambulance takes to enter the accident site and then take the victim to the hospital. Due to the continuous growth in road accidents, road safety gains considerable attention by industries and researchers. A smart accident detection and warning system is necessary to reduce the number of deaths from road accidents. It will alert all emergency services like hospitals, police stations, etc., once the system detects an accident. A number of accident detection and alert systems have been introduced in the last few years. Most of the existing approaches use the Global Positioning System (GPS) to find the vehicles' locations. Some vehicles are even equipped with GPS, sensing the location, and sending it to the cloud [6]. e-Notify is one of the popular accident alert systems requiring an on-board unit (OBU) in every vehicle [7,8]. The eCall system was developed by the European Commission and it is compulsory to deploy in each vehicle developed after 2015. When the accident has been identified by eCall, it informs the emergency number 112 (like 109 in USA, 108 in India) [9]. Because of cloud computing [10] and BigData [11–13], it is now possible to process the massive amounts of data that we receive from satellites, radar, and GPS systems [14–16]. Even in today's world, it is possible to follow the movement of each individual object using a GPS. In the past few years, Internet of Things (IoT) and deep learning-based solutions have been extensively used to develop road safety solutions for smart cities.

This paper proposes a smart accident alert system for the smart cities that includes a force sensor, GPS module, alarm controller, ESP8266 controller, camera, Raspberry Pi, and GSM module to collect and transmit accident-related information to the cloud/server. Deep learning techniques are used on the cloud to measure the severity of the accident and alert the police station and hospitals accordingly. The proposed system consists of two phases: in the first phase, an accident is detected using IoT and deep learning. In the second phase, accident information is sent to the emergency departments for the rescue operation. The contribution of the proposed work is given below:

- An IoT and AI-based integrated model is proposed.
- Ensemble transfer learning with dynamic weights is used to minimize the false detection rate.
- The Ensemble transfer learning model with dynamic weights is used to curtail the false positive rate.
- The Haversine formula is used to find the shortest distance between two points on Earth.
- The proposed method is validated by a comparative analysis of ResNet and InceptionResnetV2.
- To measure the performance of the proposed approach in the real world, it is validated on the toy car.

The rest of the paper is organized as follows: Section 2 covers some existing techniques used to detect and rescue accidents, Section 3 discuss the existing pre-trained model used

for training and testing the model, Section 4 describes the proposed architecture used for the accident detection and rescue operations using IoT and deep learning, Section 5 discuss the proposed methodology, Section 6 emphasizes the performance of the proposed architecture, and Section 7 concludes the entire work and gives future directions.

## 2. Literature Survey

The identification and alert system of road accidents is a critical task and has gained much attention from researchers. Numerous accident detection and alert methods have been proposed in the past few decades. In general, accident detection and alert methods are mainly classified into three categories:

1.  Smart Phone-Based Systems [17–21];
2.  Hardware-Based Systems [22–26];
3.  Integrated Accident Detection System [27–29].

### 2.1. Smartphone-Based Systems

In these systems, a smartphone is used to detect and notify road accidents using various smart phone sensors. Zhao et al. [17] used an accelerometer and GPS data to identify accident locations. This approach only detects an accident and does not inform any emergency services. Reddy et al. [18] proposed an accident detection method that identifies the accident location using a GPS module and notifies the hospitals by sending a message via mobile. The main limitation of this paper is that they are using only one sensor. So, if the sensor does not work, the entire system will fail. Moreover, it may also increase the false alarm.

Hamid M. et al. [19] proposed a smartphone-based accident detection and notification system. The G-force value, noise, and speed are used for detecting car accidents. The mobile sensor and accelerometer (microphone) are used for extracting gravitational strength and noise. Once the accident is detected, a message is sent to the emergency number. As speed is used as a key parameter for accident detection, this device could have a false alarm rate when the accident happens at a low speed. Patel et al. [20] have created an Android application for accident identification and notification. They have used only an accelerometer mobile sensor and GPS module to detect an accident. Once the accident is identified, it sends a pre-recorded voice message to the emergency number (108). The main limitation of this paper is that they have used only one sensor, so the entire system fails if the accelerometer sensor does not work. Isha Khot et al. [21] developed an Android app that uses only the accelerometer sensor and GPS module of a smartphone for accident detection and notification. One of the main features of this app is that it also notifies the nearby user about the accident location and the emergency services. They also rely on one sensor.

The main limitation of the smartphone-based system is the dependency on the smartphone. Each user must have a smartphone with a pre-installed accident detection app. The accuracy of the system solely depends upon the quality of the mobile sensor. It may also have a higher chance of false accident detection in the case of slow speed.

### 2.2. Hardware-Based Systems

These types of accident detection systems use various sensors for accident detection and notification. Typically, these sensors are mounted outside the car.

D. Bindu et al. [22] proposed a vehicle accident detection and prevention system which uses a vibration sensor and microcontroller. All components such as sensors, GSM module, GPS module, etc., are connected to the Atmel microcontroller AT89S52. Sensors often interpret the data such as acceleration, force, vibration, etc., and transmit the alert to all stored emergency numbers until the accident has been identified. This system does not support rescue operations. A. Shaik et al. [23] suggested an IoT-based accident alert and rescue system that uses a GPS sensor and accelerometer to gather all information and send it to the cloud. A message will send to all emergency numbers stored in the victim's profile

in the case of an accident. This message provides information about accident severity and accident location that will help the ambulance reach the accident location as soon as possible. C. Nalini et al. [24] discussed an IoT-based vehicle accident detection and rescue system. This work uses a vibration sensor, buzzer, and GPS module for accident detection and notification. When the accident has been identified, a buzzer will start for one minute, and if a buzzer is not off within one minute, a message will send to the emergency numbers through web services. C. Dashora et al. [25] introduced an IoT-based accident detection and notification system similar to that proposed in [24]. The main limitation of this paper is that once the accident is detected, the location will be sent to the customer care department, and a person will call the nearby hospital. It involves human intervention. N. K. Karanam et al. [26] proposed a fully automated accident detection and rescue system that uses a crash sensor and GPS module to develop this system.

The main limitation of the IoT-based systems is that they are an expensive solution compared with the smartphone-based system. The accuracy of the system can be improved by using some AI-based techniques.

### 2.3. Integrated Accident Detection System

These types of ADRS use a combination of IoT and deep learning for the accident detection and rescue system. Fogue et al. [27] proposed an integrated accident detection and rescue system that uses an on-board unit (OBU) to identify and notify about vehicle accidents. All accident-related information like speed, type of vehicle involved in an accident, airbag status, etc., are gathered through various sensors and sent to the cloud. Then, machine learning approaches are used to predict the accident. The main limitation of this approach is that not all vehicles support an OBU. R. Gokul et al. [26] introduced a deep learning-based accident detection and rescue system. A Wi-Fi web camera is pre-installed to the highway, which continuously monitors the vehicle passing through the road and sends it to the cloud. At the cloud, a video analysis is completed through the deep neural network. If the accident is detected, information is sent to the nearby control room. S. Ghose et al. [28] proposed a convolutional neural networks (CNN)-based accident detection system. This model has two classes, named accident and non-accident. Live video of a camera is continuously transferred to the server where CNN is used to classify the video into two categories: accident and non-accident. To train a model, they have created their dataset from the YouTube videos. The authors claim that the accuracy of the accident detection model is 95%. This system only detects accidents and does not consider the rescue operations. C. Wang et al. [29] introduced a computer vision-based accident detection and rescue system. The pre-trained Yolo v3 model was used for accident identification. The Yolo model is trained in a different situation to detect accidents like rainy, foggy, low-visibility, etc. They have used image augmentation techniques to increase the dataset size. The model's input is the live video from the camera, which is configured on the road. The author claims the accuracy of the model is 92.5% with a 7.5% false alarm rate.

Akash Bhakat et al. [30] used the combination of IoT and machine learning techniques to detect accidents and activate the rescue module. Initially, they collect the accident-related information through the IoT kit and send it to the server for processing. Instead of using the cloud or server for processing the collected data, this approach uses the fog computing to process the information locally. This method employs a machine learning methodology to detect accidents, which may not provide acceptable accuracy due to the complexity of the input video data.

Jae Gyeong Choi et al. [31] suggested an ensemble deep learning-based method to detect accidents from the dashboard camera. In this approach, video and audio data are collected from the dashboard camera, and then it uses gated recurrent unit (GRU) and CNN to detect and classify the accident. The major limitation of this approach is that the camera is located on the dashboard, which may be damaged during the accident. This approach relies only on the camera and does not use any IoT module, which may increase the false positive rate.

Hawzhin Hozhabr Pour et al. [32] introduced an automatic car accident detection approach which uses the combination of CNN and support vector machine (SVM) to detect the accident. This work uses different feature selection approaches to select the most prominent features from the available features set. The author claims the highest 85% accuracy at the testing time, which is not acceptable in the real-world applications. In addition, the IoT module is discussed in detail.

A. Comi et al. [33] applied some data mining approaches to determine the primary cause of road accidents and the pattern of those accidents. In order to conduct the analysis, the author collected data on traffic accidents that occurred between 2016 and 2019 in 15 districts of the Rome Municipality. In light of the findings, the authors assert that for the descriptive analysis, Kohonen networks and k-means algorithms are better suited, whereas neural networks and decision trees are more suited for the predictive analysis.

L. Li et al. [34] employed data mining techniques to determine the nature of the relationship between the cause of the accident and the environment in which it occurred on the FARS Fatal Accident dataset. The priori algorithm is used to establish the association rule on the attributes that are major causes for the accident, such as a drunk driver, weather condition, surface condition, light condition, collision manner, etc. Then, a k-means clustering algorithm is used to create the clusters, and finally a classification model is built using the the Naive Bayes classifier. The suggested model also gives some tips on how to drive safely. A similar task was performed by S.A. Samerei [35] on bus accidents. They have collected bus accident data over the period of 2006–2019 in the State of Victoria, Australia. A set of association rules are derived to find the major cause of bus accidents.

E. S. Part et al. [36] came up with a way to find out how fast a car was going and how often it crashed. An algorithm called "path analysis" is used to find this relation. The author draws the conclusion that the rate of travel of the vehicle has a positive correlation with the collision.

G. Singh et al. [37] proposed a simple deep learning-based model for predicting road accidents. This model uses one input layer, one output layer, and two hidden layers for the prediction purpose. The model is trained using a real-time dataset that only contains 148 samples, which is insufficient for training a deep learning model.

In the past few years, the deep learning-based solution has gained so much popularity in every sector such as medical, agriculture, and transport [38,39]. The primary requirement of all deep learning models is the huge amount of quality data during training and testing times. Most of the integrated accident detection systems use cameras that are configured on the highways. These cameras continuously monitor the traffic and send information to the cloud [40], where deep learning approaches are used to classify a video into an accident and non-accident class. The main challenge faced by the researchers is the unavailability of the accident dataset. The system's accuracy solely depends on the camera's quality and various conditions such as low-visibility, rain, fog, etc. Moreover, the maintenance of the cameras on highways is a very challenging task. Most of the work completed so far has used the deep neural network or CNN to train the model. As the sufficient amount of data are not available, a pre-trained model can be used to increase the accuracy of the accident detection model.

## 3. Pre-Trained Model

As deep learning models require a huge amount of data, the training and accuracy of the deep learning-based model depend upon the amount and quality of data. To obtain high-quality, real-time data, an IoT kit can be developed which collects the live data from the accident location and transfers it to the cloud for the further process. The pre-trained model is one of the most powerful solutions when we have a small dataset. These models are well-trained on a huge dataset with higher accuracy. Therefore, we need to train these models only for our specific task. Hence, a pre-trained model can be trained on a small dataset with higher accuracy. A number of pre-trained models are available such

as VGGNet, ResNet, InceptionNet, InceptionResNetV2, etc. In this work, we have used ResNet and InceptionResNetV2 to developed smart ADRS.

### 3.1. ResNet-50

ResNet-50 was developed by Kaiming He et al. in 2015 [41]. Instead of using a standard sequential network such as VGGNet, it is a deep neural network that is composed of many different building blocks. This model has been trained to classify images into 1000 different categories. This model has 175 convolution blocks and one classification layer with 1000 neurons and Softmax activation function. The layered architecture of ResNet-50 is illustrated in Figure 1.

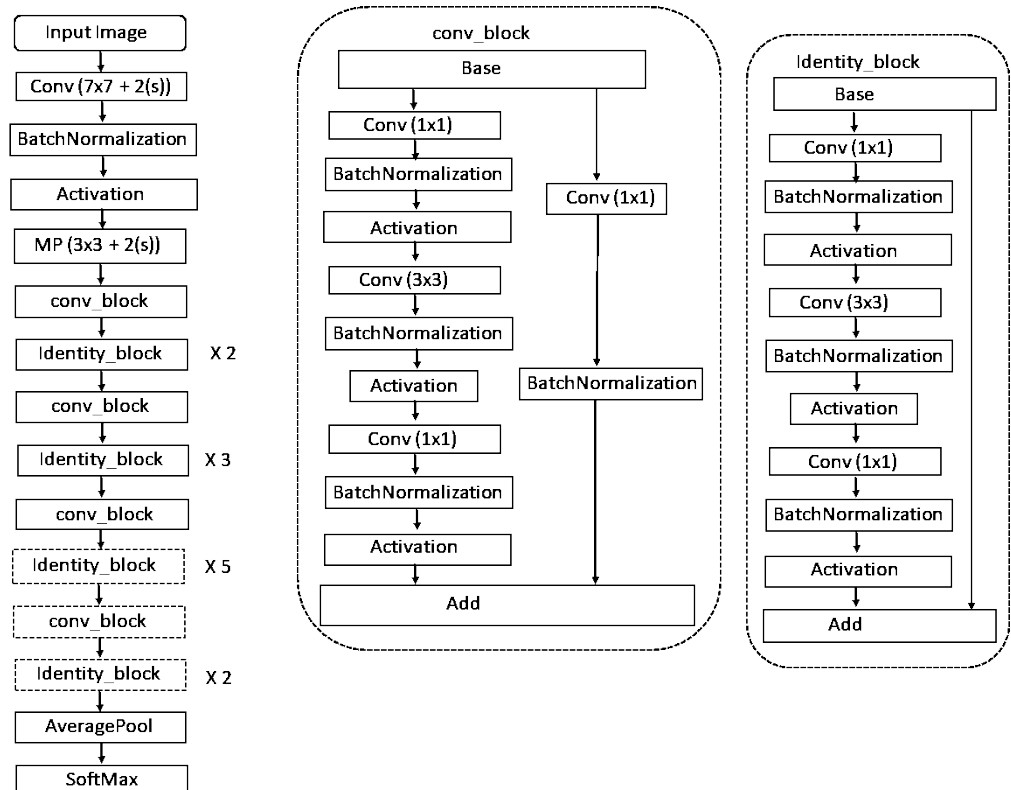

**Figure 1.** ResNet-50 layered architecture (He K. et al., 2016 [41]).

### 3.2. InceptionResNetV2

C. Szegedy et al. developed InceptionResNetV2 in 2017 [42]. This model incorporates the most advantageous characteristics of both the inception net and the ResNet (residual net). Among the most complex neural networks, it has 780 inception blocks, making it one of the most comprehensive (1x1) convolution without activation). The final classification layer is a dense layer of 1000 neurons with a Softmax activation function. InceptionResNetV2 has a layered design, which is depicted in Figure 2.

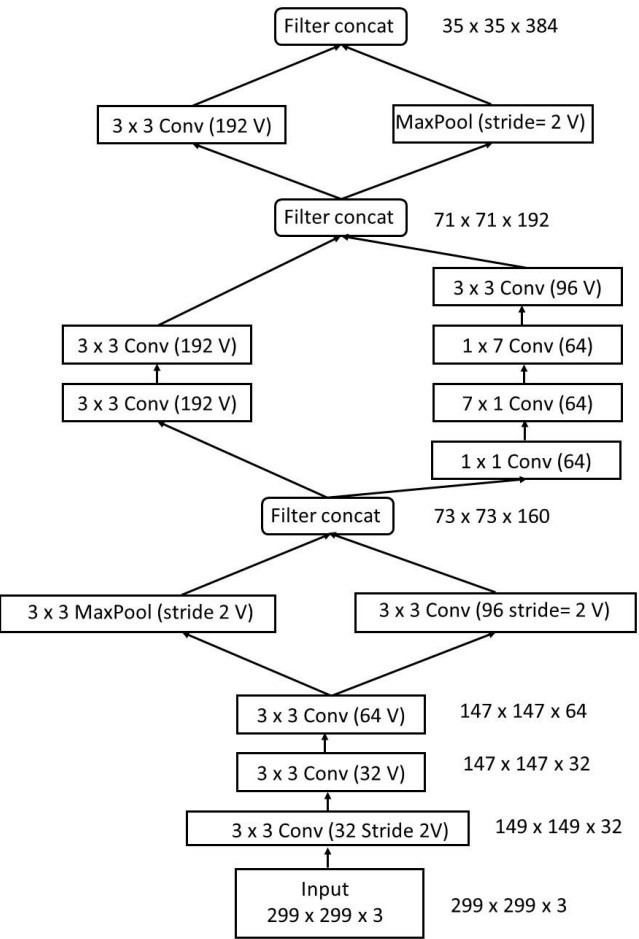

**Figure 2.** InceptionResNetV2 layered architecture (Szegedy C. et al., 2017 [42]).

## 4. Proposed Architecture

This work introduces an integrated accident detection and reporting system (ADRS) to overcome existing shortcomings in the ADRS. To achieve higher accuracy, we use a combination of IoT and deep learning. The proposed ADRS is comprised of two phases. The first phase is responsible for accident detection using IoT and deep learning, and the second phase is taking care of the rescue operations. Figure 3 illustrates the architecture of the proposed ADRS.

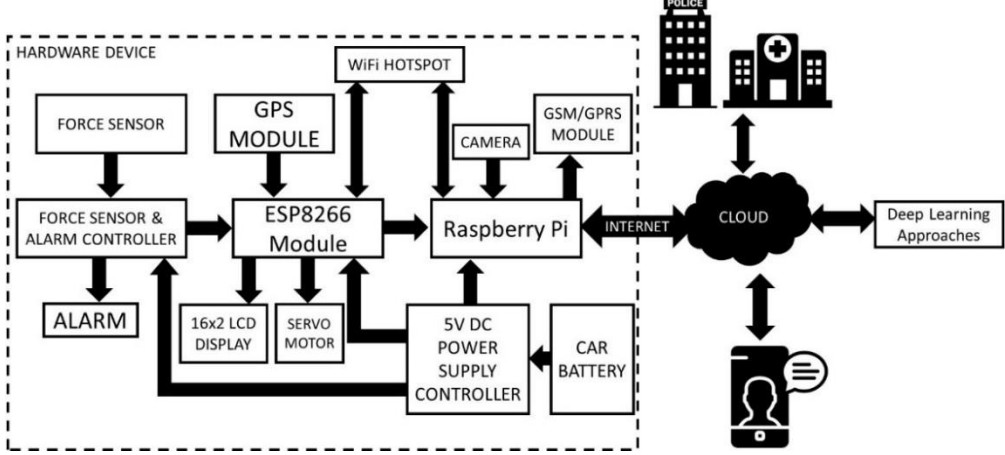

**Figure 3.** ADRS architecture.

### 4.1. Phase 1: Accident Detection

This module uses the capabilities of IoT and AI to detect an accident and its intensity. Various sensors are used to collect accident-related information such as location, force, etc., and identify accidents. In order to minimize the false alarm rate, deep learning techniques are used. The details of the IoT and AI module are as follows:

### 4.1.1. IoT Module

This module is the main module of the proposed ADRS and primarily responsible for accident detection. It uses various sensors and other IoT devices. The entire hardware system requires a 5V DC power supply connected to the vehicle battery and a separate power bank to provide power to the system. A detailed description of various sensors and other components is given below:

**Force Sensor:** It is the main component of the ADRS. We are using the FlexiForce A401 sensor to measure the impact (force) on the car [43]. It has a 2-pin connector with the largest 25.4 mm sensing area. This sensor is available in a single force range varying from 111 N (0–25 lb). The driving voltage and resistance of the feedback resistor can be adjusted to alter the operating range.

**ATMEGA328:** It is a microcontroller unit that controls various devices and is directly connected to the force sensor [44]. If the value of force is more than 350 Pa, then an accident is detected [27]. Once the accident is detected, it triggers the alarm, and if the alarm is not reset within 30 s, it will send the signal to activate NodeMCU.

**NodeMCU- ESP8266:** It is a master microcontroller unit that is connected to the ATMEGA328 microcontroller, Raspberry Pi, GPS module, and servo motor. It is a Wi-Fi-enabled controller that transmits all accident-related information to the cloud [45].

**Raspberry Pi:** Raspberry is a tiny computer system with OS, and it performs all tasks that a desktop/laptop can achieve [46]. It is a powerful controller used to connect IoT devices. In our ADRS, Raspberry Pi 3B+ is connected to a camera, which sends the videos and photos to the cloud to measure the accident's severity.

**Pi Camera:** The Pi camera is mounted on the servo motor and connected to the Raspberry Pi [47]. Once the accident is detected, it will take the video and photos and send them to the cloud, where deep learning is used to analyze these videos and pictures to measure the severity of the accident. Deep learning is mainly used to minimize the false alarm rate and measure the intensity of the accident.

**Servo Motor:** It is connected to the NodeMCU, which rotates the camera from 0 to 180 degrees [48].

**GSM SIM900A:** It helps in the rescue operation and sends messages to the various emergency services like a police station, hospitals, relatives, etc. It is connected to the Raspberry Pi [49,50].

**GPS Neo-6m Module:** It is used to find the accident location and is connected to the NodeMCU.

**LCD Display:** One 16 X 2 LCD is used to display the accident location captured by the GPS module.

### 4.1.2. Deep Learning Module

Deep learning algorithms have been greatly improved over the last few decades and may now be used in any domain with high accuracy [49–53]. Most of the existing ADRS just identify the accidents either using IoT sensors or deep learning. As a result, as soon as an accident is identified, information is transmitted to all emergency numbers. These approaches may have a higher false detection rate because they depend entirely on the sensor. So, if the user stops their car suddenly, the existing solutions detect it as an accident. To minimize the false detection rate, we are using a transfer learning-based pre-trained model named ResNet and InceptionResnetV2. These models are trained to classify input video into two classes, i.e., accident or not accident. If the model's output is accident, then only the accident location is shared with the emergency services.

### 4.2. Phase 2: Notification

Once the accident is detected, an alarm will start for 30 s. If the vehicle's passenger resets the alarm, only nearby mechanics' information will be sent to the registered mobile. It will help the driver if the vehicle has any technical issues. In contrast, if the alarm is not reset after 30 s, then the location information is immediately sent to the nearby hospitals and police station. This phase uses various databases to store the useful information.

Databases

The proposed ADRS uses four databases to manage the entire system. Detailed information of the different databases is given below.

**User Personal Details Database:** This database contains all user-related information like vehicle owner, address, and relative's contact number.

**Vehicle Database:** This database comprises all details of the vehicle like Vehicle_Number, Vehicle _Name, and Vehicle_Type.

**Hospital Database:** The hospital database stores all information on the hospitals.

**Police Station Database:** It stores all the information about the police station along with nearby hospital details.

**Mechanic Database:** This database contains all information regarding the mechanics. Each mechanic needs to install an Android app through the Google Play store and fill in all details.

## 5. Proposed Methodology

The proposed system uses a combination of the Internet of Things and deep learning to develop an ADRS.

### 5.1. IoT Module for Accident Detection

The designed hardware device contains sensors and actuators to detect an accident and its intensity. A force sensor is used to identify the accident which is mounted on the vehicle chassis. The force sensor is connected to the ATMEGA328 microcontroller to detect accidents and raise the alarm system. If the speed and force's values exceed a predefined threshold, $T_{speed}$ and $T_{force}$, an accident will occur. When the accident is detected, it will first activate the alarm for 30 s. If there is no accident or the accident is average or negligible, the driver can reset the force sensor and alarm controller by simply pressing a button. NodeMCU will send the nearby mechanics' information to the registered mobile that can be used to resolve any technical issue in the vehicle. If the button is not pressed before 30 s, it will send a legal accident signal to the ESP8266 module. The ESP8266 module will activate the camera and send video, photos, GPS data, and other items to the Master Controller Raspberry Pi. The GPS data are also sent to the LCD display to show GPS information captured by the GPS module [51,52]. Pseudocode for accident detection is shown in algorithm 1.

Once the accident is detected, longitude and latitude details are sent to the cloud and the rescue module. The rescue module is responsible for sending the accident details to all emergency services. All the information of the vehicle owner, relatives, and mechanics can be retrieved from the database. However, we need to find the nearest hospital and police station [23,25]. To find the nearby hospital and police station, we are using the Haversine formula, which determines the shortest path between two points when they are on Earth. This formula is mainly used to find the shortest distance between two locations when you know the navigation latitudes and longitudes. The Haversine formula can be expressed by using the following Equations (1)–(4):

$$haversine(\theta) = sin^2\left(\frac{\theta}{2}\right) \tag{1}$$

$$\left(\frac{d}{r}\right) = haversine(\varnothing_2 - \varnothing_1) + \cos(\varnothing_1)\,\cos(\varnothing_2)\,haversine(\gamma_2 - \gamma_1) \tag{2}$$

$$d = rhav^{-1}(h) = 2rsin^{-1}\left(\sqrt{h}\right) \tag{3}$$

$$d = 2rsin^{-1}\left(\sqrt{sin^2\left(\frac{\varnothing_2 - \varnothing_1}{2}\right) + \cos(\varnothing_1)\cos(\varnothing_2)sin^2\left(\frac{\gamma_2 - \gamma_1}{2}\right)}\right) \tag{4}$$

where $r$ = 6371 km represents Earth's radius; the distance between points is d; $\varnothing_1$, $\varnothing_2$ are the latitudes of the two points; and $\lambda_1$, $\lambda_2$ are the longitudes of the two points, respectively.

---

**Algorithm 1:** For Accident Detection

---

Input Data: Value of force (F) and speed (S)
Output: Accident status
    acc $\leftarrow$ 0
    **if** (F > $T_{force}$ & S > $T_{speed}$) **then** acc $\leftarrow$ 1
    **else if** (F > $T_{force}$ | S > $T_{speed}$)
        acc $\leftarrow$ 1
    **end if**
    **if** acc= 1 **then**
        Activate alarm and set alarm timer (AT = 0)
        Alarm_OFF_timer $\leftarrow$ Alarm_OFF()
        **If** (AT >= 30 seconds) **then**
            status= Accident_detected
        **else**
            status= no_accident
            Get nearby mechanics details from database and send message to the owner
        **end if**
    **end if**
    **if** (status= Accident_detected) **then**
        final_status = call Deep_Learning_Module()
        **if** (final_status= Accident_detected) **then**
            Get accident location from GPS
            call rescue_operation_module()
    **else**
        Get nearby mechanics details from database and send message to the owner

---

**Algorithm 2:** For Rescue Operation

---

    Input Data: longitude and latitude
    Output: Inform to all emergency services
        Slat = starting latitude
        elat = end latitude
        s1on = starting longitude
        e1on = end longitude
        dis_lat = elat – slat
        dis_lon = elon - slon
        Dist = R * H
        Find nearby police station and hospital using Haversine
        Hostp = nearby_hospital
        Police_station = nearby_police_station
        Get car details, vehicle details, mechanic details
        Send message to cloud and all emergency services using GSM module

---

The ADRS can connect to the cloud by Wi-Fi present in a Raspberry Pi and ESP8266 using HTTP and MQTT protocols, respectively. Figure 4 shows the flow diagram of the proposed system.

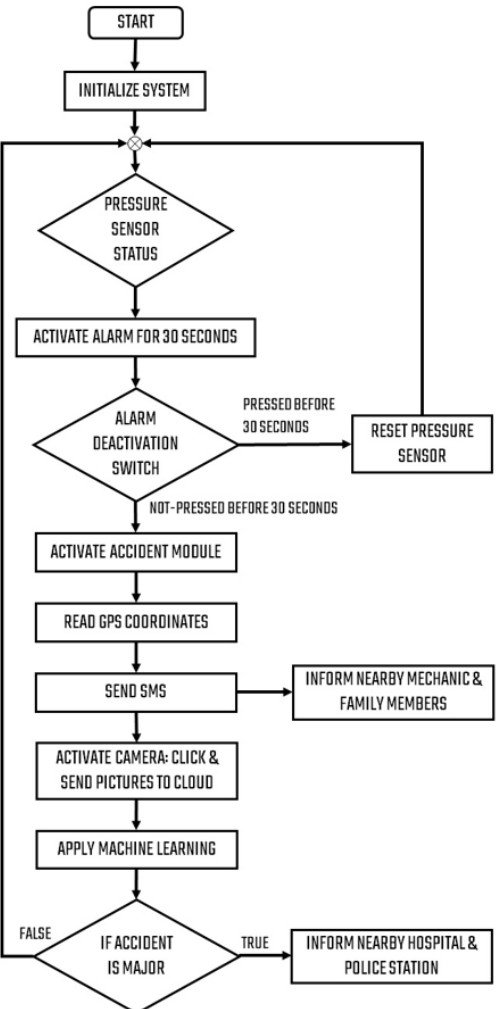

**Figure 4.** Flow diagram of the proposed ADRS.

*5.2. Deep Learning Module for Accident Detection*

The main goal of this module is to cut down on the number of false alarms. One of the main causes of road accidents is the speed of vehicles on the road. Most of the existing ADRS mainly identify an accident based on the speed sensor. If the vehicle's speed is changed suddenly and exceeds the predefined threshold, then the system detects an accident. So, the false detection rate of these systems can be high, because in various circumstances, the vehicle's speed can be changed, such as a speed breaker, an obstacle on the road, a technical problem of the vehicle, etc. The proposed ADRS uses the deep learning technique to minimize this misidentification rate, which takes the input video from the dashboard's camera.

Once the device has detected the accident, NodeMCU will activate the camera and give the activation signal to Raspberry Pi. The camera will take some images, record the 30 s video, and upload it to the cloud for analysis. In the cloud, a pre-trained ResNet and InceptionResnetV2 model is trained to identify current situations as accidents or not accidents. First, the input video is converted into the frames and passes to the model. This module will help us to minimize false accident deletion and improve the accuracy of the model. The deep learning module architecture is depicted in Figure 5.

Two steps are involved in the deep learning module of the proposed ADRS. These steps are:

1. Dataset;
2. Customization and Training of the Pre-Trained Model.

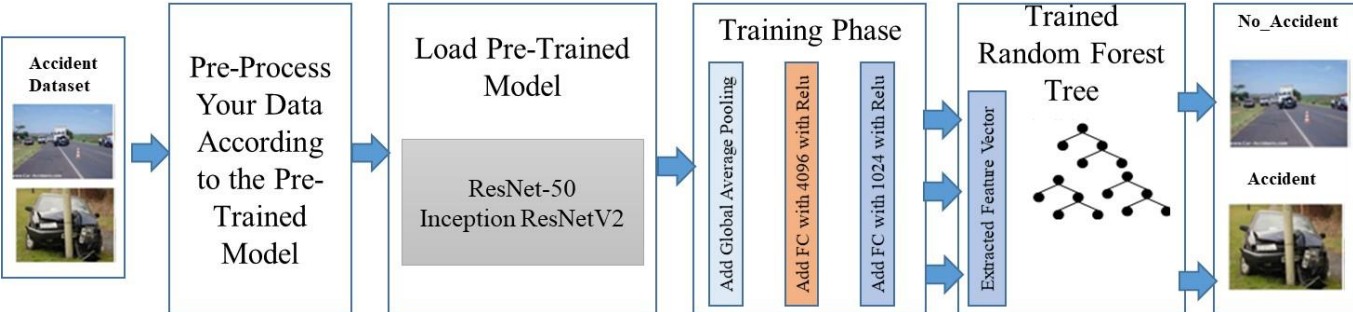

**Figure 5.** Deep learning architecture of proposed ADRS.

5.2.1. Dataset

The quality and amount of the dataset have the greatest influence on the performance of the model. Since the standard dataset is not available, we built our dataset from the YouTube videos. We downloaded many YouTube videos and converted them into frames using the CV2 Python library. The entire dataset is split into two parts: training and testing. Training data are used to train the model, while test data are used to measure the model's performance at test time. As shown in Table 1, training and test data are balanced for both the classes to avoid overfitting. The number of images in our dataset is 500, with 250 images in each class (accident, not accident). Image augmentation techniques are frequently used to increase the size of a dataset, which generates more numbers of images by applying various operations on the images such as rotation, zoom out, zoom in, sharping, etc. The entire dataset is divided into two parts: training (used to train model) and testing (used to evaluate model performance). Table 1 provides an in-depth summary of the dataset.

**Table 1.** Dataset description.

|  | No. of Accident Images | No. of Not Accident Images |
| --- | --- | --- |
| Total Images | 250 | 250 |
| Train | 205 | 205 |
| Test | 45 | 45 |

5.2.2. Customization of the Pre-Trained Model

We are employing a pre-trained model due to the limited amount of data available. In addition to learning hidden patterns from the huge dataset, the pre-trained models have already been trained on a large dataset. In this work, we have trained two well-known pre-trained models called ResNet and InceptionResnetV2. Both the pre-trained models have been trained on large datasets which include 1.2 million images and are capable of categorizing an image into 1000 different categories. This dataset is being made available as part of a competition named the ImageNet Large Scale Visual Recognition Challenge (ILSVRC) [21]. All pre-trained models are composed of a large number of poling layers (PL) and convolution layers (CL). Feature extraction is completed by CL, whereas the pooling layer is responsible for coping with the size of the image and avoiding overfitting it. If the size of the input image and kernel is represented by I(n ×m) and K(a, b), respectively, then the two-dimensional convolution operation can be represented by Equation (5).

$$S_{ij} = (I * K)_{ij} = \sum_{a=0}^{n-1} \sum_{b=0}^{m-1} I_{i+a,j+b} * K_{a,b} \tag{5}$$

where $S_{ij}$ represents the $i$th and $j$th pixel value of an image.

It is necessary to inject nonlinearity into the model at each convolution layer, and this is accomplished through the use of an activation function. The activation functions, namely, ReLu, LeakyReLu, and Softmax are the most popular activation functions in pre-trained models. The mathematical equation for various activation functions and pooling operations is shown in Table 2.

**Table 2.** Different CNN operations and its mathematical equation.

| Operation | Equation |
|---|---|
| Max Pooling | $out_i = max_{1<j<aXb}\left(x_j\right)$ |
| Avg. Pooling | $out_i = avg_{1<j<aXb}\left(x_j\right)$ |
| ReLu | $ReLU(z_i) = max(0, \; z_i)$ |
| LeakyReLu | $LeakyReLU(z_i) = max(0.01z_i, \; z_i)$ |
| Sigmoid | $\sigma(z_i) = \frac{1}{1+e^{-Z_i}}$ |
| Softmax | $softmax(z_i) = \frac{e^{Z_i}}{\sum_j e^{Z_j}}$ |

where the output of *i*th pooling layer is represented by $out_i$; output of layer *i*th pre-activation function is represented by $Z_i$; and (a, b) represent the filter's size. As the traditional ResNet and InceptionResnetV2 are trained to classify 1000 different objects, our ADRS has only two classes, so we need to customize these models. To begin, we will delete the pre-trained model's final layer (the classification layer) and a global average pooling layer (GAPL) is added in their place [22]. The GAPL calculates an input dimension by taking the mean of all features and converting an input dimension of $h \times w \times D$ to $1 \times 1 \times D$. The output of the GAPL is distributed to the five dense layers, each of which contains 2048 neuronal connections and a LeakyReLu activation function. After five fully connected layers, one classification layer with one neuron is added, which performs the classification task. The random forest tree classification algorithm is used in the classification layer to train the model. The main aim of using this classifier is to add monitoring because the deep learning model is a black box. It is an ensemble method in which numerous trees are generated and a prediction is made based on a majority vote among the trees.

The pseudocode for the random forest decision tree is shown in algorithm 3 where k represents the classifier's number of tress; F shows a number of features in the dataset; N and L represent the internal nodes and leaf nodes, respectively.

---

**Algorithm 3:** Random Forest Tree

---

(1) Divide the dataset into training and test set
(2) Set the value of number of tree m

    2.1 Build m number of trees say $(R_1, R_2, R_3 -------R_m)$
    2.2 Train model for the training data based on Gini index and entropy.
    2.3 Without purring, extend each tree to its furthest possible length.

(3) To evaluate the model performance pass new instance X

    3.1 For new instance X, using all k trees $(R_1, R_2, R_3 -------R_m)$, predict the class of an instance X.
    3.2 Use voting approach to finally predict the class on X.

---

The input of the random forest decision tree (RFDT) is the one-dimension vector that is given by the fifth fully connected layer. Suppose that the input to the RFDT is the feature vector $(\vec{A}, \; B)$, where the input feature vector is represented by $\vec{A}$ and the corresponding actual value is represented by Y. After the training, for each input image, the model will find the set of vectors which can be expressed by Equations (6) and (7),

$$X = \left\{ (A^1, \; Y^1), \; (A^2, \; Y^2), \; (A^3, \; Y^3) \; --------- \; (A^n, \; Y^n) \right\} \tag{6}$$

where *n* represents the number of sample images in the input vector.

$$X\prime = \left\{ (F_1^1, \; Y^1), \; (F_2^2, \; Y^2), \; (F_3^3, \; Y^3) \; -- \; (F_m^m, \; Y^m) \; --- \; (F_n^n, \; Y^n) \right\} \tag{7}$$

where the feature vector of the *m*th input image and corresponding target label are represented by $F_m^m$ and $Y^n$, respectively. Predict the output $Y'$ by feeding the pre-trained model's extracted features $X'$ to the random forest classifier. The absolute difference between the actual and anticipated values is referred to as loss or error, and it can be calculated using Equation (8).

$$\text{Loss or error} = L(||Y - Y'||) \tag{8}$$

where Y and $Y'$ represent the actual and predicted value; loss function is denoted by L(x). The Bayesian cross-entropy loss function (BCEF) is used in our proposed model which can be expressed by Equation (9).

$$L(W) = -\frac{1}{n} \sum_{i=1}^{n} log(f(x_i, c_i, W)) \tag{9}$$

Using this loss function, all weights are changed in the network, and the modified weights are propagated back to the network. As an optimizer function, the stochastic gradient descent optimizer is used, which is expressed by Equation (10).

$$W = W - \sigma . \nabla_W J\left(W, x^{(i)}, y^{(i)}\right) \tag{10}$$

The different symbolic notations and their definitions are shown in Table 3.

**Table 3.** Symbolic notation and its meaning.

| W | Weight |
|---|---|
| $\sigma$ | Learning Rate |
| $\nabla_W$ | Partial Derivation |
| $x^{(i)}$ | Response to the Input |
| $x^{(i)}$ | Response to the Output |

The Algorithm 4 for the deep learning module of the proposed ADRS is given below.

| **Algorithm 4:** Deep Learning for Accident Detection |
|---|
| Input: Training dataset which is collection of X-ray images Data= {$(A^1, Y^1)$, $(A^2, Y^2)$, $(A^3, Y^3)$ ——$(A^n, Y^n)$} |
| Parameter: Initialize all the parameters like weight W, learning rate $\sigma$, batch size, threshold $\varnothing$, max_iteration. |
| Output: Accident, No_Accident |
|  o Perform pre-processing like augmentation, rescaling, etc., to the input data D<br> o Customize the pre-trained model by removing top layers<br> o First add GAPL to flatten the network and then add convolutional layers<br> o Configure five dense layers after CL where each layer has 2048 neurons<br> o To avoid overfitting, dropout 20% neurons from each dense layer<br> o Perform feature extraction from the dense layer and store all extracted features into the feature vector ($D^{feature}$).<br>  o Size of $D^{feature}$ Will be 1x 2048 for one image. So, for n image it will nx2048<br> o Create data frame of size<br>  o $D^{feature} =${$(F_1^1, Y^1)$, $(F_2^2, Y^2)$, $(F_3^3, Y^3)$ $- - - - - (F_n^n, Y^n)$}<br> o Initialize RFT with k number of trees (i.e., k=50)<br> o Train the RFT model for the extracted features $D^{feature}$<br> o Compile model for binary cross-entropy loss function and SGD optimizer |

## 6. Experiment Setup

The ADRS consists of two modules, namely, accident detection and rescue system. As the performance of the proposed ADRS cannot be evaluated into the actual vehicle, we have placed a developed IoT kit to the toy car. All components such as the sensors, alarm,

controller, etc., are mounted on the car. The GPS module is used to measure the vehicle speed and accident location. Figure 6 shows the setup of various devices into the car.

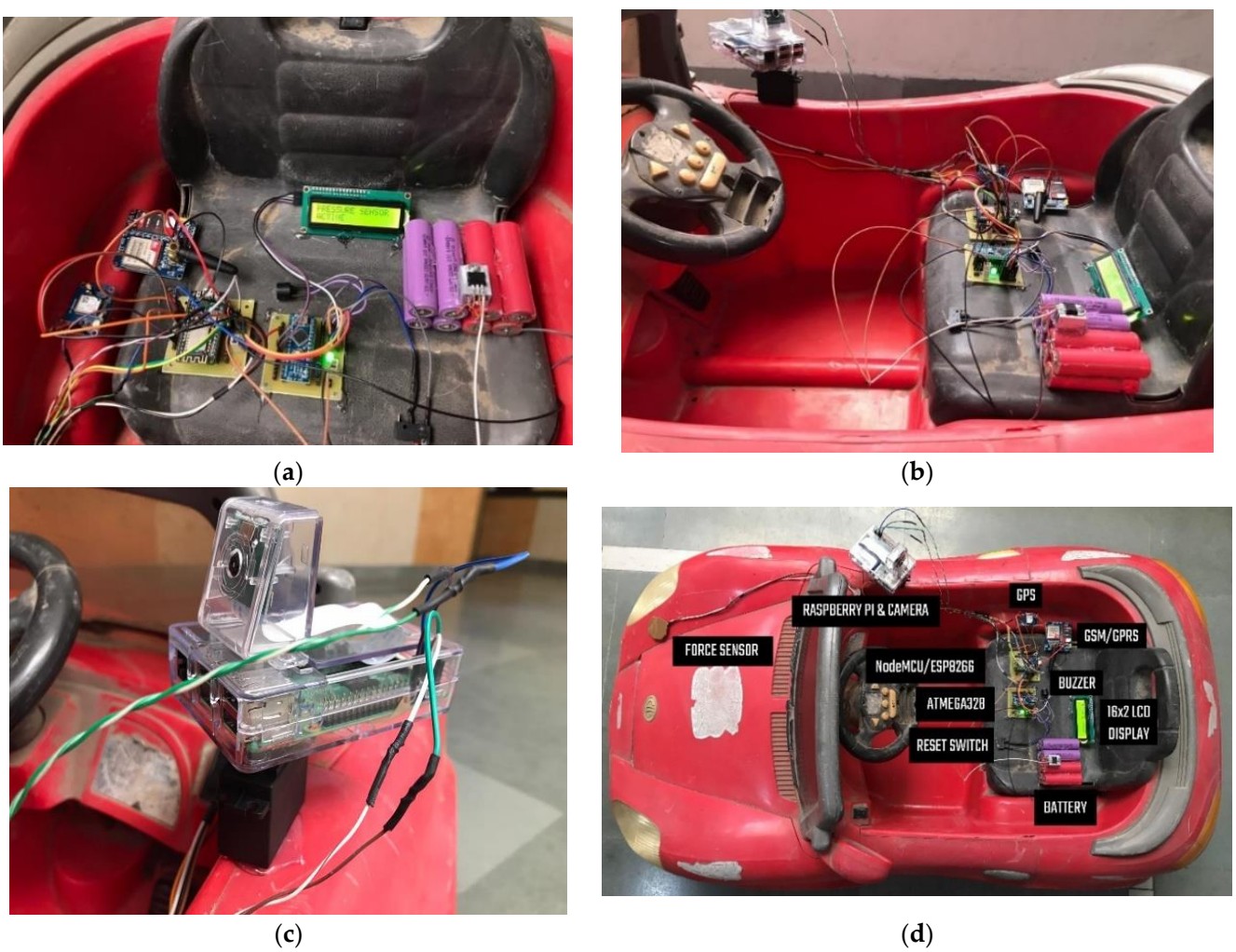

**Figure 6.** Installation of developed IoT kit on a car. (**a**) Car with IoT Sensor and LED Display, (**b**) Car with IoT Sensor, (**c**) Car with IoT Kit with Camera, (**d**) Car with labelling of each Device in IoT Development kit.

The confusion matrix, precision, recall, and area under the curve (AUC) are all metrics that are used to measure the performance of the deep learning-based model [49,50]. The formulas for different evaluation metrics are given by Equations (11)–(15).

$$\text{Accuracy (A)} = \frac{TP + TN}{TP + TN + FP + FN} \qquad (11)$$

$$\text{Precision (P)} = \frac{TP}{TP + FP} \qquad (12)$$

$$\text{Recall (R)} = \frac{TP}{TP + FN} \qquad (13)$$

$$\text{F1 Score (F1)} = 2 \ast \frac{Precision \ast Recall}{Precision + Recall} \qquad (14)$$

$$\text{AUC} = \int_a^b f(x).d(x) \qquad (15)$$

Here, TP denotes the true positive rate, FP denotes the false positive rate, TN denotes the true negative rate, and FP represents the false positive rate.

To simulate the proposed deep learning module, Python programming is used and implemented in Google Colab. All training and test images are resized to 224 × 224 in ResNet-50 and 299 × 299 in Inception ResNetV2. A detailed description of the simulation environment is shown in Table 4.

**Table 4.** Training Setup.

| | |
|---|---|
| No. of Iterations | 50 |
| Batch Size | 64 |
| Learning Rate | 0.001 |
| Multiprocessing | False |
| Shuffle | True |

### 6.1. ResNet-50

After removing the classification layer from the traditional ResNet-50, the remaining total layers are 175. Then, we have added five layers to the model. As a result, the improved ResNet-50 has a total of 180 layers. Due to the fact that 175 layers have already been trained, only the newly added five layers require additional training. Figure 7 depicts the confusion matrix for the improved ResNet model during the training and testing phases.

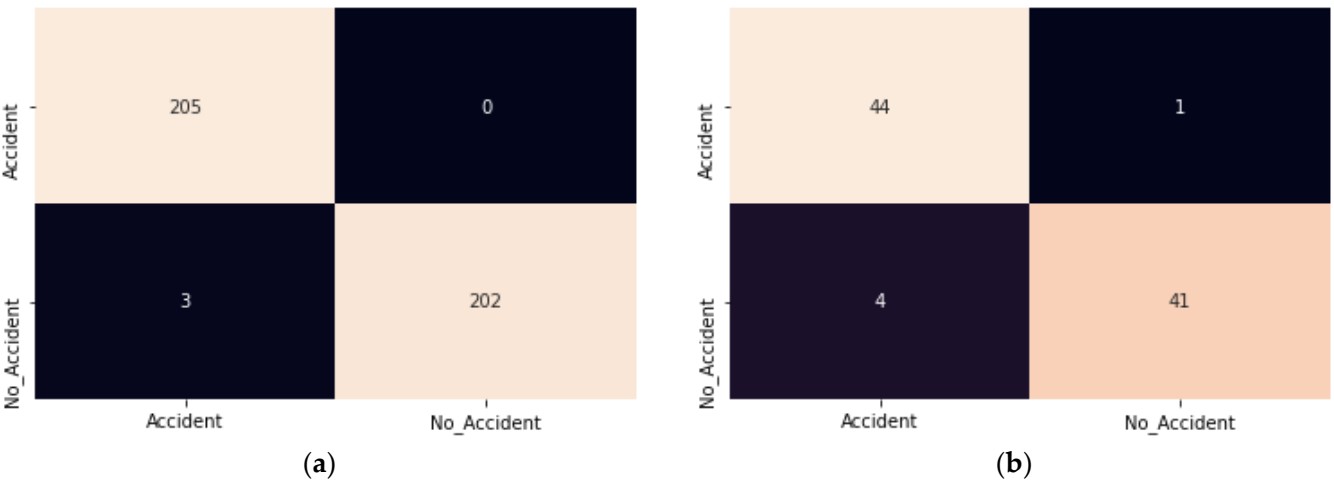

(**a**)          (**b**)

**Figure 7.** Confusion matrix for ResNet-50 (**a**) training data; (**b**) test data.

It is observed in Figure 7a that false positive (FP) in the modified ResNet model is minimum approx. zero. It indicates that in none of the cases is the accident class classified as a no_ accident. The accuracy and loss results for 50 of the training and test iterations shown in Figures 8 and 9.

After analyzing Figures 8 and 9, the proposed model can learn the hidden pattern in the input image in less than 50 iterations. Tables 5 and 6 illustrate the classification report of the training and test phase.

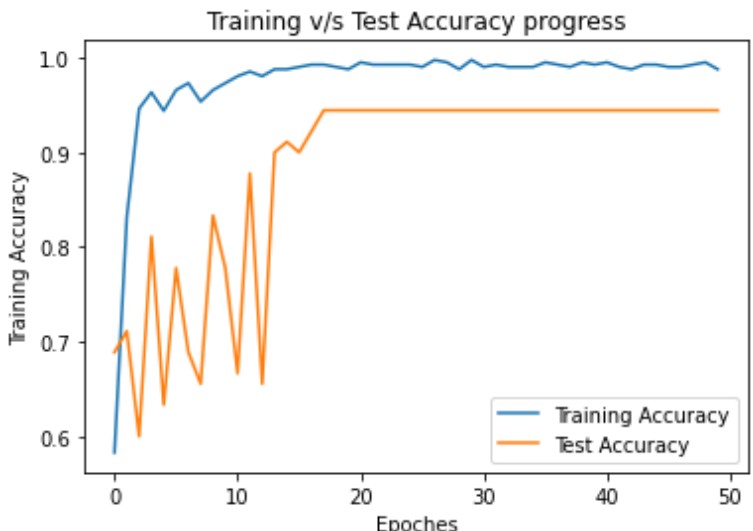

**Figure 8.** Training v/s test accuracy for ResNet-50.

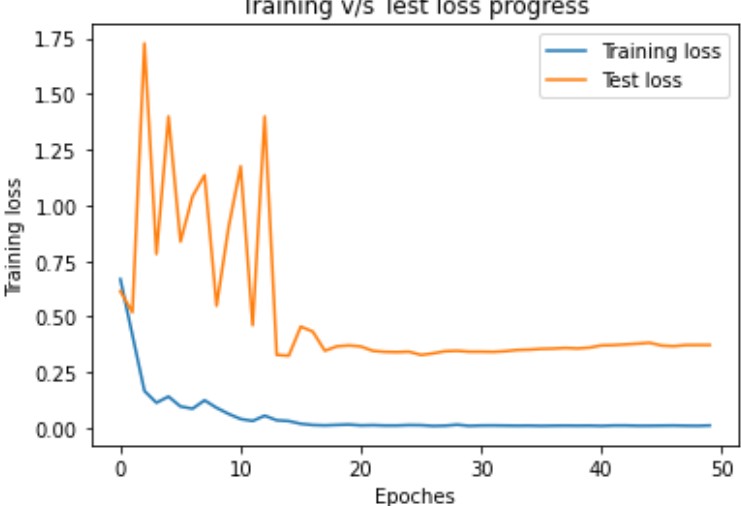

**Figure 9.** Training v/s test loss for ResNet-50.

**Table 5.** ResNet-50 classification report for training phase.

|  | P | R | F1 | S (Support) |
|---|---|---|---|---|
| Accident | 0.99 | 1.00 | 0.99 | 205 |
| No_Accident | 1.00 | 0.99 | 0.99 | 205 |
| A |  |  | 0.99 | 410 |
| macro avg | 0.99 | 0.99 | 0.99 | 410 |
| weighted avg | 0.99 | 0.99 | 0.99 | 410 |

**Table 6.** ResNet-50 classification report for testing phase.

|  | P | R | F1 | S |
|---|---|---|---|---|
| Accident | 0.92 | 0.98 | 0.95 | 45 |
| No_Accident | 0.98 | 0.91 | 0.94 | 45 |
| A |  |  | 0.94 | 90 |
| macro avg | 0.95 | 0.94 | 0.94 | 90 |
| weighted avg | 0.95 | 0.94 | 0.94 | 90 |

The modified ResNet-50 model achieves the training and test accuracy of 99.3 and 99.4, respectively. The main benefits of this approach are the recall rate. The accident recall rate of the model is 1. That refers to all accidents detected as an accident. Figure 10 shows the area under the curve (AUC) for the modified ResNet model.

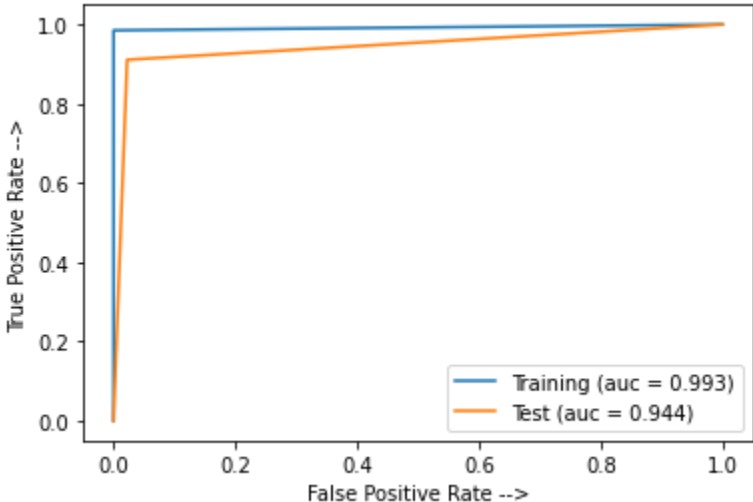

**Figure 10.** AUC for ResNet.

Figure 11 shows the classification of the accident image during the training and test phase, respectively.

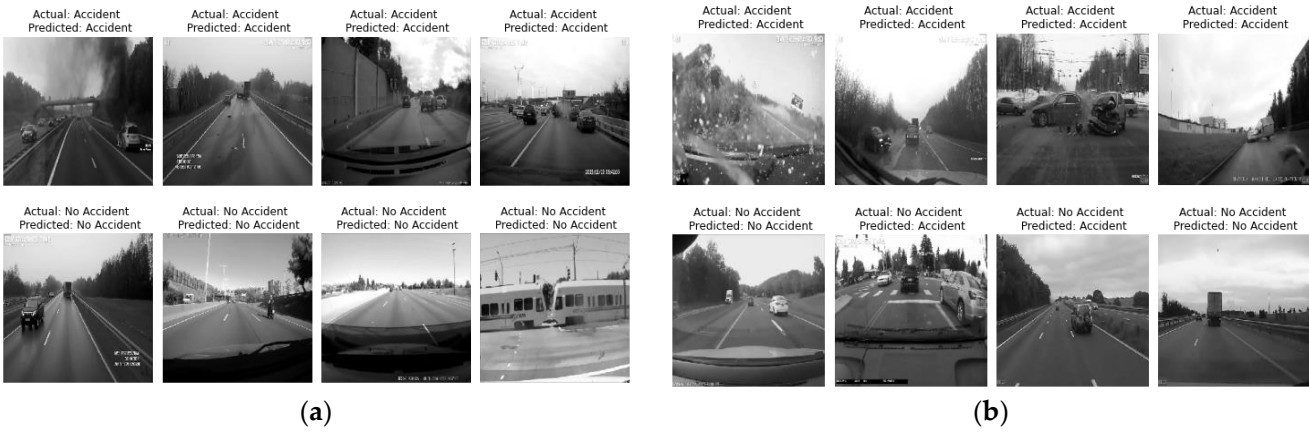

(**a**)                                                          (**b**)

**Figure 11.** Image classification during training and test phase. (**a**) Training phase; (**b**) test phase.

### 6.2. InceptionResNetV2

After updating the traditional InceptionResNetV2, the total number of layers in the updated InceptionResNetV2 is 785, where we need to train only the newly added five layers. The confusion matrix for the training and testing phases is shown in Figure 12.

It is observed in Figure 12 that this model classified some accident classes as a no_accident class. An effective ADRS must have a minimum false positive rate if possible. Figures 13 and 14 show the training and test accuracy and loss, respectively, for the 50 iterations.

Tables 7 and 8 show the classification report for the trained, validation, and testing phases.

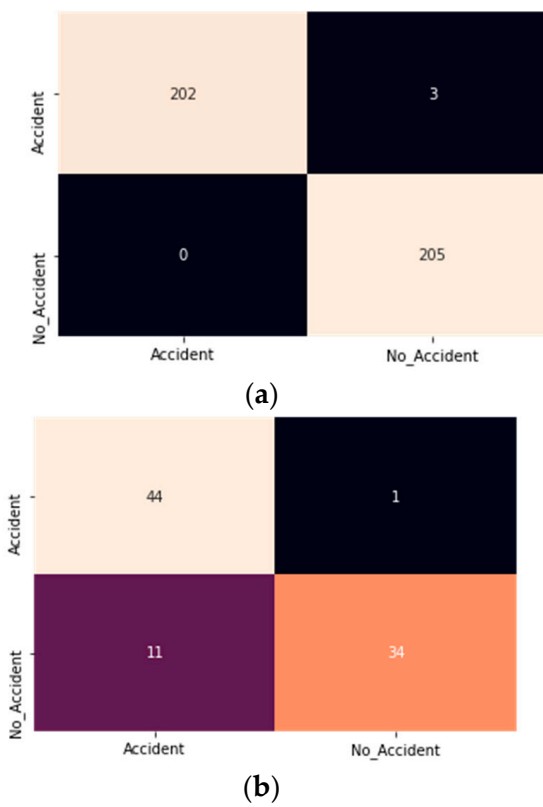

**Figure 12.** Confusion matrix for (**a**) training data; (**b**) test data.

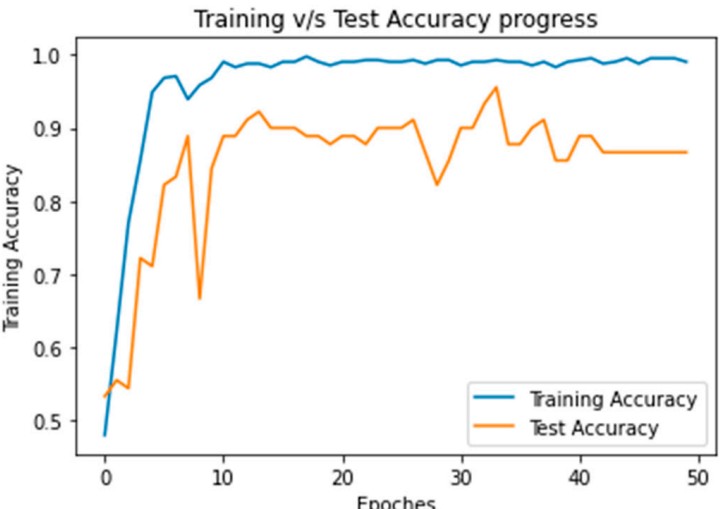

**Figure 13.** Training v/s test accuracy for InceptionResNetV2.

**Table 7.** InceptionResNetV2 classification report for training phase.

|  | P | R | F1 | S |
|---|---|---|---|---|
| Accident | 1.00 | 0.99 | 0.99 | 205 |
| No_Accident | 0.99 | 1.00 | 0.99 | 205 |
| A |  |  | 0.99 | 410 |
| macro avg | 0.99 | 0.99 | 0.99 | 410 |
| weighted avg | 0.99 | 0.99 | 0.99 | 410 |

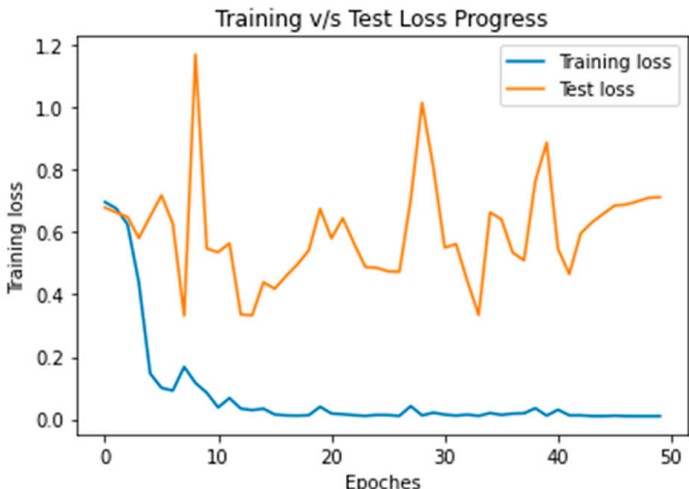

**Figure 14.** Training v/s test loss for InceptionResNetV2.

**Table 8.** InceptionResNetV2 classification report for testing phase.

|  | P | R | F1 | S |
|---|---|---|---|---|
| Accident | 0.80 | 0.98 | 0.88 | 45 |
| No_Accident | 0.97 | 0.76 | 0.85 | 45 |
| A |  |  | 0.87 | 90 |
| macro avg | 0.89 | 0.87 | 0.86 | 90 |
| weighted avg | 0.89 | 0.87 | 0.87 | 90 |

The training and test accuracy of the model is 99.3% and 86%, respectively. Hence, the modified InceptionResNetV2 is overfitted and not appropriate for the ADRS. Figure 15 shows the AUC for the modified InceptionResNetV2.

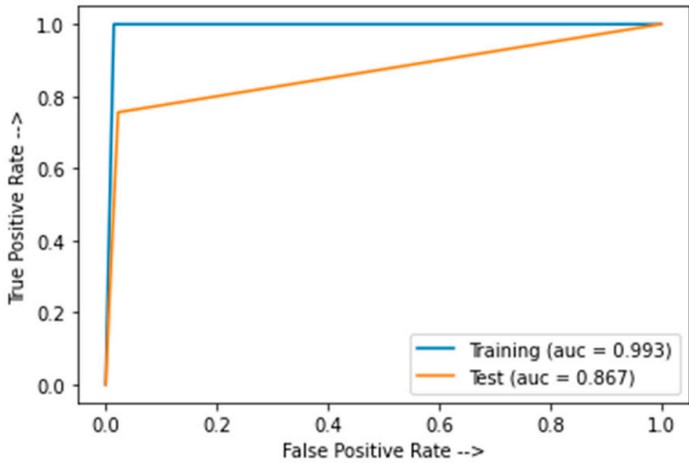

**Figure 15.** AUC for InceptionResNetV2.

Figure 16 shows the classification of the accident image during the training and test phases.

Comparative studies of ResNet-50 and InceptionResNetV2 are shown in Figure 17. Both the models give the same training accuracy (99.3%), but ResNet-50 has more test accuracy than the InceptionResNetV2. Furthermore, ResNet-50 has an almost 100% recall rate of accident class. It means there is less probability of an accident class being classified into a no accident class. An efficient ADRS should have a minimum false positive rate; therefore, ResNet-50 is a better choice for developing an accident detection model compared to the InceptionResNetV2.

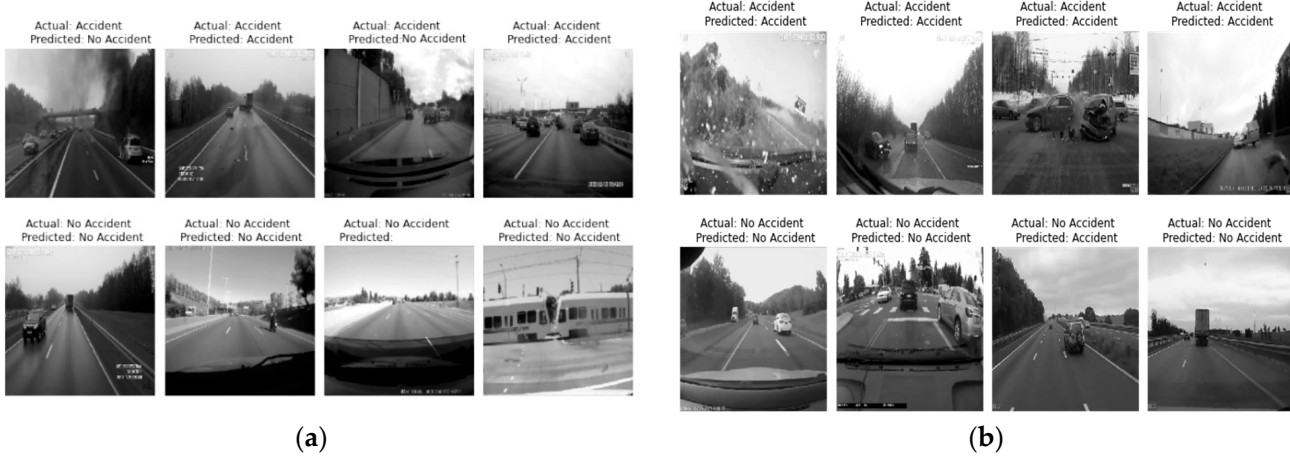

**(a)**                                                                                     **(b)**

**Figure 16.** Image classification during (**a**) training phase; (**b**) test phase.

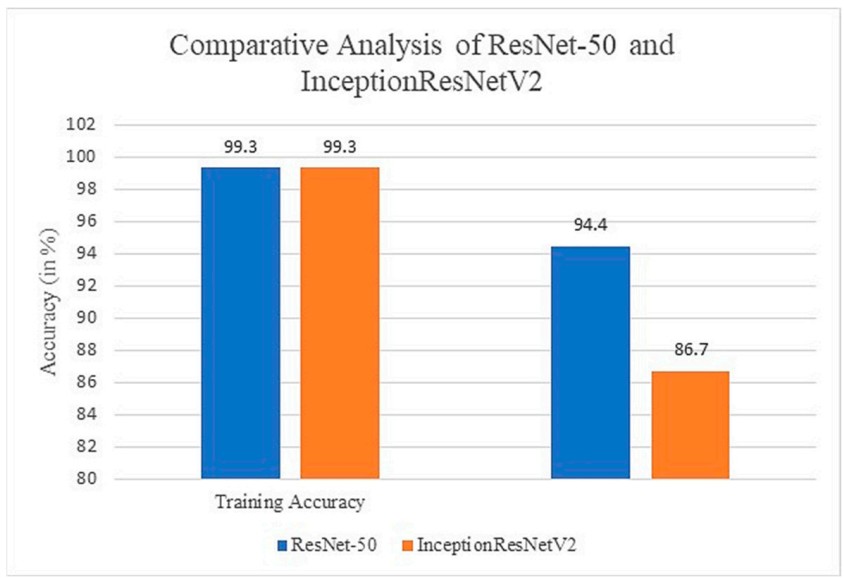

**Figure 17.** Comparative analysis of modified ResNet-50 and InceptionResNetV2.

## 7. Conclusions and Future Work

In the smart city, due to the good quality of the roads, drivers run their vehicles at high speeds, resulting in an increase in road accidents. Although a range of accident detection and prevention systems are being launched on the market, many fatalities still occur. At least some of the issue is exacerbated by an insufficient automatic identification of accidents, ineffective warning, and emergency service responses. This work is carried out in two phases: in the first phase, an IoT kit is used for the accident detection, and then a deep learning-based model is used to validate the output of the IoT model and perform the rescue operation. Once an accident has been identified by the IoT module which uses a force sensor to measure the impact on the car and a GPS module for the vehicle speed, it transfers all useful information to the cloud. In the second phase, pre-trained models, namely VGGNet and InceptionResNetV2, are used to minimize the false detection rate and activate the rescue module. If an accident is detected by the deep learning module, a rescue module is activated and the details are sent to the nearest police station, hospitals, and relatives. The Haversine formula is used to find the shortest distance between two points. Experiment results show that ResNet-50 is a more suitable model for accident detection than the InceptionResNetV2, as it has a higher test accuracy and recall rate of the accident class. To measure the performance of the proposed model in the real world, the model is implemented on a toy car.

The proposed model can help us to minimize the death rate due to the unavailability of emergency services at the accident location. Due to the combination of IoT and AI, the model has zero false positives during the training time, and extremely low false positives at the test time. The proposed model does not consider the security aspect, so we intend to address this issue in future work. In addition, some driver alert systems such as the drowsiness detection module can also be added in the proposed model.

**Author Contributions:** N.P. conceived and performed the research and formal analysis, help in paper drafting; R.K.G. decide the methodology and wrote the paper; Y.S. analyzed the data and perform the visualization; A.S. perform the data curation and help in implementation. M.M. and M.B. provided funding support and guidance and wrote revised paper. All authors have read and agreed to the published version of the manuscript.

**Funding:** The authors would like to thank the Taif University Researchers Supporting Project, number (TURSP-2020/239), for the support. Taif University, Taif, Saudi Arabia.

**Institutional Review Board Statement:** Not Applicable.

**Informed Consent Statement:** Not Applicable.

**Data Availability Statement:** All data is publicly available and information is already shared in the manuscript.

**Acknowledgments:** The authors would like to thank the Taif University Researchers Supporting Project, number (TURSP-2020/239), for the support. Taif University, Taif, Saudi Arabia.

**Conflicts of Interest:** The authors have no conflict of interest.

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
