# Peer review of "AI Enabled Accident Detection and Alert System Using IoT and Deep Learning for Smart Cities"

_sustainability, doi:10.3390/su14137701_

Round 1

Author Response

The authors greatly appreciate the valuable and insightful comments made by the reviewers and would like to thank them for their efforts. Their comments have undoubtedly helped us to improve the quality of our manuscript. We have carefully considered all the comments and the manuscript have been revised thoroughly to meet their expectations. Our point-by-point responses to each comment are listed in the attached file.

Thank you

Reviewer 2 Report

The article is interesting but requires a strong revision in terms of a scientific problem, and purpose. A literature review can hardly be called a literature review, but rather a random selection of matching works by other authors. The methodological part is interesting, but it should also be refined, in particular in justifying the selection of methods (why exactly this way and with this measure). The described formulas should be explained. There is also no information on testing the data for the desired properties, which are included in the assumptions of a given method (the data should be verified - their distributions before applying a given method).

Conclusions and discussions of the results are too meager. The authors focused very much on the technical description of their system, rather than on the scientific merits of this operation. This is not an objection, but the article should be adapted to the requirements for scientific articles (subject your thoughts and research results to strict rules).

Abbreviations should not be used in the title, but in their full wording. Note applies to AI and IoT.

The abstract does not contain key information, it is more suitable for the introduction than for the compact description of the article.

The keywords were spelled correctly, although I believe they should be sorted alphabetically. One keyword has an improper abbreviation (ITS - Intelligent Transportation Systems instead of Transportation Systems). IoT has been doubled. Moreover, they should be full wording instead of abbreviations.

In the introduction on the first page, the statistics are given without the source of the data.

The introduction formulates a goal, but it is not correctly formulated. In addition, the word "suggested" is used instead of "projected". It should be reformulated and indicate the purpose of this system (that is, indicate what it contributes to, and what is the purpose of using AI in this context).

On page 2, there is an unfinished sentence, "The Internet of Things (IoT) and Deep Learning have

recently been extensively used to develop road safety solutions in smart cities. With the ".

The description of the article structure should include all sections, including the introduction.

In section 2, please provide references to the methods listed on page 2 (separate references for each method - after the name of the method).

The literature review should be refined, these are sentences taken out of context, which does not form a logical whole, although they are contained in one paragraph. It is difficult to understand what this literature review brings to the research (what is its value). There is no link between the literature review and the rest of the manuscript. The authors must add a few sentences justifying their argument/research problem with the support of similar statements in the literature. The same goes for methods. What is the conclusion that Author X used this method and author Y used a different method for this research?

Other:

The list of references must be corrected, it contains glaring errors.

The remaining statements at the end of the article are missing.

Figure 1 and Figure 2 are illegible (poor quality).

Author Response

(The authors gave the same response as above.)

Reviewer 3 Report

Review report: AI-Enabled Accident Detection and Alert System using IoT and Deep Learning for Smart Cities.

The paper is interesting and the final aim of the contribution is clear. The paper is well-written and well-structured enough.
I appreciate the effort the authors made.

. The scientific content of this paper is correct. 

. The results are well presented.

. The technical quality of this paper is correct. 

. The conclusion is correctly justified and supported by the results. 

Quality of presentation:

. The abstract should be extended and better written. 

. Better highlight novelty in the study.

. Better define motivations for the research.

. Add limitations of the presented approach.

Scientific soundness :

. The subject addressed in this paper is relevant.  

Interest to the readers :

. In my opinion, the methodology of this paper seems to be interesting for the readership of the journal.

Author Response

(The authors gave the same response as above.)

Round 2

Reviewer 2 Report

Substantial revisions were made to the manuscript where necessary. The authors' responses are exhaustive. I have no substantive reservations.

Congratulations.

Only please pay attention to double enters / white spaces between paragraphs) /

and on the reference list - the citing style needs to be slightly improved to MDPI, also eg 36 & 37 - should combine into one; 38 & 39; 41 & 42 and next with www (in one line is the title of www, the next – URL).

Author Response

The authors greatly appreciate the valuable and insightful comments made by the reviewers and would like to thank them for their efforts. Their comments have undoubtedly helped us to improve the quality of our manuscript. We have carefully considered all the comments and the manuscript have been revised thoroughly to meet their expectations. Our point-by-point responses to each comment are discussed in the attached file.
